# Seroprevalence and effect of HBV and HCV co-infections on the immuno-virologic responses of adult HIV-infected persons on anti-retroviral therapy

**Lawrence Annison** [1,2]*, **Henry Hackman** [2], **Paulina Franklin Eshun** [1], **Sharon Annison** [3], **Peter Forson** [1], **Samuel Antwi-Baffour** [4]

1 Department of Medical Laboratory Technology, School of Allied Health Sciences, Narh-Bita College, Tema, Ghana, 2 Department of Medical Laboratory Technology, Faculty of Applied Sciences, Accra Technical University, Accra, Ghana, 3 Department of Epidemiology and Disease Control, School of Public Health, University of Ghana, Legon, Accra, Ghana, 4 Department of Medical Laboratory Sciences, School of Allied Health Sciences, College of Health Sciences, University of Ghana, Korle-Bu, Accra, Ghana

⊚ These authors contributed equally to this work.

* lannison@atu.edu.gh

**Data Availability Statement:** All relevant data are within the paper and Supporting Information files.

## Abstract

Chronic hepatitis negatively affects persons living with HIV. While varying in their transmission efficiency, HIV, HBV, and HCV have shared routes of transmission. Available data suggest widely variable rates of HBV and HCV infections in HIV-infected populations across sub-Saharan Africa. With prolonged survival rates due to increased accessibility to antiretroviral drugs, HBV and HCV have the potential to complicate the prognosis of HIV co-infected patients by contributing significantly to continued morbidity and mortality. The study sought to determine the seroprevalence of HIV/HBV and HIV/HCV co-infections among HIV patients on antiretroviral therapy and to evaluate the effect of HIV/HBV and HIV/HCV co-infections on the immunologic and virologic responses of patients. A cross-sectional study in which samples were taken from 500 people living with HIV and attending ART clinic at the Fevers unit of the Korle Bu Teaching Hospital and tested for Hepatitis B Surface Antigen (HBsAg) and Hepatitis C virus antibody (HCV). CD4 cell counts and HIV-1 RNA levels were estimated as well. Data generated were analysed using IBM SPSS version 22. The seroprevalence of HIV/HBV and HIV/HCV co-infections among people living with HIV was 8.4% and 0.2% respectively. HIV/HBV coinfection included 15/42 (35.7%) males and 27/42 (64.3%) females out of which the majority (97.6%) were in the 21–60 years old bracket. HIV/HBV and HIV/HCV co-infections have varied effects on the immunological and virological response of HIV patients on ART. The mean CD cell count was 361.0 ± 284.0 in HIV/HBV co-infected patients and 473.8 ± 326.7 in HIV mono-infected patients. The mean HIV-1 RNA level was not significantly different ($X^2$ [df] = .057 [1]; $P$ = .811) among HIV/HBV co-infected patients ($Log_{10}$2.9±2.0 copies/mL), compared to that of HIV mono-infected patients ($Log_{10}$2.8±2.1 copies/mL) although HIV mono-infected patients had lower viral load levels. One-third (14/42) of HIV/HBV co-infected patients had virologic failure and the only HIV/HCV co-infected patient showed viral suppression. 336/500 (67.2%) patients had HIV-1

**Funding:** The author(s) received no specific funding for this work.

**Competing interests:** The authors have declared that no competing interests exist.

viral suppression (females [66.1%]; males [33.9%]) while 164/500 (32.8%) had virologic failure (females [67.7%]; males [32.3%]). The mean CD4 count of patients with viral suppression and patients with virologic failure was 541.2 cells/µL (95% CI 508.5–573.8) and 309.9 cell/µL (95% CI 261.9–357.9) respectively.The study concludes that, HIV/HBV and HIV/HCV coinfections do not significantly affect the immunologic and virologic responses of patients who have initiated highly active antiretroviral therapy, and treatment outcomes were better in females than in males. There was no HBV/HCV co-infection among patients.

## Introduction

HIV continues to be a major public health concern as many more people are infected daily [1]. Currently, about 38 million people are living with the disease globally with 1.7 million newly infected people as at the end of 2018 [2]. Human Immunodeficiency Virus (HIV) attacks the body's immune system, specifically the CD4+ T cells, which help the immune system fight off infections. Untreated, HIV reduces the number of CD4+ T cells in the body, making the person more likely to get other infections or HIV-related cancers [3]. Over time, HIV destroys many of these cells leading to a weakened immune system (acquired immunodeficiency syndrome (AIDS)). This results in the body not being able to fight off other diseases and then opportunistic infections take advantage of the body's weak immune system [4].

Hepatitis B Virus (HBV) and Hepatitis C Virus (HCV) are two notable causes of chronic and severe forms of viral hepatitis [5] that have emerged as important public health issues globally, characterized by high prevalence, high morbidity and high mortality [6, 7]. Globally, an estimated 296 million people are living with hepatitis B virus infection (defined as hepatitis B surface antigen positive) and an estimated 58 million people have chronic hepatitis C infection [8, 9].

Chronic hepatitis disproportionately affects persons living with HIV [10]. In sub-Saharan Africa, which bears the highest burden of infectious diseases and remains the epicentre of the global HIV epidemic [11, 12], HIV, HBV and HCV are believed to be transmitted independently but can have shared routes of infection and co-infection is very common. HBV and HCV infections are transmitted by parenteral routes [13]. HIV is heterosexually acquired during adulthood. The interaction between HIV and HCV co-infection affects the transmission and natural history of HCV infection [14, 15].

HIV/AIDS remains a manageable chronic disease since there is no cure. HIV mono-infected patients respond favourably to HAART, but when co-infected with viral hepatitis, the response is suboptimal [16]. HBV-associated liver diseases are a major cause of morbidity and mortality in people living with HIV (PLHIV) [17]. Thus, HBV and HCV have the potential to complicate the prognosis of HIV co-infected patients [18]. HIV infection exacerbates the natural progression of HBV and HCV, and as a result, increases the risk of cirrhosis and end-stage liver disease in HBV and HCV co-infection [19] and promotes faster progression to chronicity, liver fibrosis, and malignancy [20]. In fact, the risk of liver-related mortality has been found to be higher (14%) in HIV/HBV co-infected patients than in HIV-mono-infected patients (6%) [21]. HBV co-infection with HIV increases the progression to HIV/AIDS and hepatotoxicity associated with the intake of antiretroviral drugs [21]. Again, in HIV-HBV/HCV co-infected patients, there is a rapid devastating effect on their liver resulting in various abnormalities, especially among the lower CD4 group [22].

Despite this, there is little data on HIV/HBV and HIV/HCV in sub-Saharan Africa due to limited diagnostic capacity [20]. In sub-Saharan Africa–which has only 12% of the global population–there were about 25.6 million (67%) HIV patients [23] in 2021 and an estimated 81 million chronic HBV carriers in 2019 [8]. Available data suggest widely variable rates of HBV (0–28.4%) and HCV (0–55.9%) infections in HIV-infected populations across sub-Saharan Africa [10, 20]. In Ghana, the HIV epidemic is generalized with a year-on-year prevalence rate consistently above 1% and the prevalence of HBV infection in the adult population is estimated to be 17.3% [24]. In Ghana, HBV/HIV and HCV/HIV coinfections are estimated to be 13.5% and 3%, respectively [25, 26].

Due to the current use of highly active antiretroviral therapy (HAART), people living with HIV (PLHIV) are living longer and AIDS–related deaths are steadily declining, especially in countries with longer treatment experience [27]. With the increased use and accessibility of HAART among HIV positive patients in Ghana, co-infection with HBV and HCV could contribute significantly to continuing morbidity and mortality among this group of patients [28]. HIV patients co-infected with HBV and HCV are unable to recover immunologically [16]. Although, this occurrence varies widely from country to country and region to region [29], and despite the public health concerns, very few studies have reported on the impact of HIV and viral hepatitis co-infection in Africa in general and Ghana in particular. Although HBV, HCV and HIV viral infections are considered to be endemic in Africa, the prevalence of HIV/HCV and HIV/HBV co-infections varies depending on the risk factors, individual behaviours, socio-demographic profiles and HBV immunization coverage [30].

Currently in Ghana, routine laboratory detection of HBV and HCV among HIV positive patients is simply non-existent, owing to the unavailability of testing materials at the treatment centers and the high cost of testing even if testing materials are available. Moreover, there is a dearth of systematic surveys on the prevalence of HBV and HCV among HIV positive patients in Ghana. Some studies have also suggested that HIV/HBV and HIV/HCV co-infections accelerate the immunologic and virologic progression of HIV patients [31]. The extent of the problem, therefore, remains unclear due to the paucity of published data to facilitate evaluation of the situation in the country. This study, therefore, sought to contribute to addressing the problem by determining the seroprevalence of Hepatitis B and Hepatitis C infection in HIV positive patients, and evaluate the effect of HIV/HBV and HIV/HCV co-infections on the immunologic and virologic response of HIV positive patients on antiretroviral therapy at the Korle-Bu Teaching Hospital, Ghana.

## Material and methods

### Study site

The study took place in the Fevers Unit of Korle-Bu Teaching Hospital, the largest and leading hospital in Ghana with a bed capacity of 2000 which serves as the major referral centre in the country.

### Study population and design

The study population included all adult HIV infected patients receiving treatment at the Fevers Unit of the Korle-Bu Teaching Hospital and were routinely scheduled for 1, 2 or 3 monthly follow–up visits. These patients had been on treatment for a mean duration of 24 months. Antiretroviral therapy was based on a triple drug regimen mainly consisting of Tenofovir, Lamivudine, and Efavirenz as the first line drug of choice. Some patients were given Dolutegravir Tenofovir, and Lamivudine as a second option, with others receiving Abacavir, Lamivudine, and Dolutegravir/Efavirenz, and a few patients receiving Combivir and Dolutegravir.

The study was a cross-sectional study and used a convenience sampling technique, in which participants were selected based on their availability and willingness to take part in the study. This was done until the study period was over. Data was collected from 500 study participants from November 2018 to February, 2019.

## Eligibility criteria

In order to be eligible, study participants were expected to meet the following criteria: Adult (18 years of age or more), documented HIV positive status, in regular HIV care follow-up at Korle-Bu Teaching Hospital, and willing and able to provide written informed consent. Participants who did not meet the inclusion criteria mentioned above were excluded from the study.

## Patient's demographic data collection

Patient's demographic data were collected using a questionnaire and entered into a Microsoft Excel (2016) sheet. This included patient's name, age, sex, and identification number from the ART clinic.

## Sample collection

Five millilitres (5mL) of venous blood sample of each participant (clinically diagnosed HIV-1 patient) was collected and two millilitre (2mL) was put in Ethylene Diamine Tetra Acetic Acid (EDTA) tube for the estimation of CD4 T cell count and HIV viral load, and three millilitre (3mL) was put in Gel Separator tube for Hepatitis B surface antigen (HBsAg) and HCV antibody serological testing.

## Hepatitis B surface antigen detection

The Advanced Quality One Step HBsAg Test (InTec PRODUCTS INC, China) was used for the qualitative detection of HBV surface antigen (HBsAg) in human serum. The test strip was immersed into the serum and then removed after 8–10 seconds and the strip laid flat on a clean surface. The presence of HBsAg was detected within 15 minutes according to the manufacturer's instructions.

## Hepatitis C antibody detection

The Advanced Quality One Step HCV Antibody Test (InTec PRODUCTS INC, China) was used for the qualitative detection of hepatitis C antibody in human serum following the manufacturer's instructions.

## CD4[+] T-cell enumeration using flow cytometry

CD4[+] T cell enumeration was done by immune-labelling and fluorescence-activated cell sorter (FACS) analysis carried out using BD FACSCount flow cytometer (Becton Dickinson, San Jose, CA, USA). The test used fluorochrome-labelled anti-CD3, CD4 and CD8 monoclonal antibody detection system. The monoclonal antibodies were contained in ready-to use reagent test vials. In addition, the reagent contained a known number of fluorochrome-integrated reference beads which served both as a fluorescence standard for locating the lymphocytes, and as a quantitation standard for enumerating the cells. Reagent test vials were brought to room temperature and vortexed upside down and then upright for 5 seconds to ensure even mixing. 50 μl of well mixed anticoagulated whole blood was added to the reagent in the test vial using the back pipetting technique with an automatic pipette and vortexed for 5 seconds. This was to allow the flourochrome-labelled monoclonal antibodies in the reagent bind specifically to

lymphocyte surface antigens. The sample was then incubated in the dark for one hour at room temperature (20–25˚C). A 50 μl fixative (5% formaldehyde in phosphate buffered saline) was added to the sample after incubation and vortexed to ensure even mixing. The sample was then run on the BD FACSCount automated reader for cell count using flow cytometry. Results of the CD4/CD3 (helper/inducer T-lymphocytes) cell counts were then printed automatically after each sample run.

### Hiv-1 viral load estimation

COBAS® AmpliPrep/COBAS® TaqMan 48 Analyzer (Roche Molecular Systems, USA), a nucleic acid amplification test system, were used for the detection and quantitation of Human Immunodeficiency Virus Type 1 (HIV-1) RNA in human plasma. Automated COBAS Ampli-Prep Instrument was used for specimen preparation whereas COBAS TaqMan 48 Analyzer was used for amplification and detection, following the manufacturer's instruction. The limit of detection was <20 copies/Ml.

### Statistical analysis

Data generated was coded, cleaned and entered into a computer for analysis. Analysis was done using IBM® SPSS® Statistics Version 22.0. Univariate analysis in the form of frequencies and percentages were performed on the demographic characteristics of the participants. Bivariate and multivariate comparisons were made between the dependent and independent variables using chi square and logistic regression respectively. This was to determine associations and test for strength of association between the dependent and independent variables. Associations were considered significant at the 95% confidence interval with p value set at <0.05.

### Ethical clearance

Ethical approval was given by the Ethics and Protocol Review Committee of the School of Allied Health Sciences, Narh-Bita College, and the management of Korle-Bu Teaching Hospital for the study to start. Informed written consent was also obtained from study participants after it was explained to them that they could voluntarily participate and withdraw without the quality of management of their conditions being compromised. Participants were also assured that the study would not adversely affect their medical condition but help improve the care of HIV patients with hepatitis B or C co-infection. The participants were further assured that any information they provided would be kept private and confidential.

## Results

### Patients' demographic and clinical characteristics

From the data obtained, there were 500 adult HIV-positive patients with complete records of HBV and HCV infection status, HIV viral load, and CD4$^+$T cell counts. The age of participants ranged from 18 to 71 years with a mean age of 43.6 ± 11.4. Age was stratified into six categories: ≤20, 21–30, 31–40, 41–50, 51–60, and ≥61 years old. The majority of participants, 154 (30.9%), were within the age group of 41–50 years, followed by those within the age 31–40 years, 116 (23.0%), 51–60 years, 114 (22.8%), 21–30 years, 51 (10%), ≥61 years, 40 (8%), and ≤20 years, 25 (5%). Again, out of the 500 participants, 333 (66.6%) and 167 (33.4%) were females and males, respectively (Table 1).

Hepatitis B surface antigen and Hepatitis C antibody (anti-HCV) were tested for in all participants and reported as reactive or non-reactive. Reactive participants were positive for either HBsAg or anti-HCV while non-reactive participants were negative for HBsAg and anti-HCV.

**Table 1. Patient demography and clinical baseline characteristics.**

| | Demography | HIV Monoinfection | HIV/HBV Coinfection | HIV/HCV Coinfection | Mean CD4 (95% CI) | P-value | Viral Load±SD | P-value |
|---|---|---|---|---|---|---|---|---|
| Characteristics | N (%) | N(%) | N (%) | N (%) | cells/uL | | log10(copies/mL) | |
| Gender | | | | | | 0.155 | | 0.1405 |
| Female | 333 (66.6) | 306 (61.2) | 27 (5.4) | 1 (0.2) | 479.0 (443.5–514.5) | | 2.82 ± 2.11 | |
| Male | 167 (33.4) | 151 (30.2) | 15 (3) | 0 (0) | 435.2 (387.3–483.1) | | 2.79 ± 2.05 | |
| **Total** | **500 (100)** | **457 (91.4)** | **42 (8.4)** | **1 (0.2)** | **464.4 (435.8–492.9)** | | **2.81 ± 2.09** | |
| Age (years) | | | | | | | | |
| ≤20 | 25(5) | 24 (4.8) | 0 (0) | 0 (0) | 489.5 (366.1–613.0) | | 3.63 ± 1.75 | |
| 21–30 | 51(10.2) | 43 (8.6) | 8 (1.6) | 0 (0) | 430.0 (352.3–507.8) | | 2.98 ± 2.30 | |
| 31–40 | 116(23.2) | 104 (20.8) | 12 (2.4) | 0 (0) | 462.6 (402.5–522.6) | | 3.08 ± 1.55 | |
| 41–50 | 154(30.8) | 142 (28.4) | 12 (2.4) | 0 (0) | 469.3 (416.2–522.5) | | 2.67 ± 2.19 | |
| 51–60 | 114(22.8) | 105 (21.0) | 9 (1.8) | 1 (0.2) | 487.0 (421.7–552.3) | | 2.64 ± 2.07 | |
| ≥61 | 40(8) | 39 (7.8) | 1 (0.2) | 0 (0) | 415.5 (325.7–505.3) | | 2.51 ± 2.01 | |
| **Total** | **500 (100)** | **457 (91.4)** | **42 (8.4)** | **1 (0.2)** | **464.4 (435.8–492.9)** | | **2.81 ± 2.09** | |

N: Number of participants

* Average age (years) 43.6 ± 11.4, minimum (18), maximum (71)

Subsequently, 42/500 (8.4% [95%CI: 6.2%-11.1%]) were reactive to HBsAg. Out of the HBsAg positive patients, 15/42 (35.7%) were males. Also, the majority (97.6%) of HBsAg positive patients were between the ages of 21 to 60 years. Only 1/500 (0.2% [95%CI: 0.0%-0.9%]) patient, a female within the 51–60 years' category, was positive for anti-HCV out of the patients tested. There was no HBV/HCV/HIV co-infections among study participants (Table 1).

## HIV-1 RNA levels

Indication of viral suppression was defined as viral load <1000 copies/mL, virologic failure as ≥1000 copies/mL according to WHO guidelines [1], detectable levels as >20 copies/mL, and levels of undetection as <20 copies/mL per the detection limit of the COBAS® TaqMan 48 Analyzer (Roche Molecular Systems, USA). Out of the patients tested, 266/500 (53.2%) had HIV-1 RNA below levels of detection (<20 copies/mL) while 234/500 (46.8%) had HIV-1 RNA detection level greater than 20 copies/Ml (Table 2). Patients with viral load below undetectable levels were made up of 177/266 (66.5%) females and 89/266 (33.5%) males. 336/500 (67.2%) patients had HIV-1 viral suppression (females [66.1%]; males [33.9%]) while 164/500 (32.8%) had virologic failure (females [67.7%]; males [32.3%]) (Table 2). Mean viral load was 2.81Log$_{10}$ copies/mL (Table 1).

## CD4$^+$ T-cell lymphocyte count

CD4 cell counts were categorized into three: Low CD4 count (<300 cells/µL), moderate CD4 count (300–700 cells/µL), and high CD4 count (≥700 cells/µL). Out of the 500 participants, 170 (34.0%) had low CD4 count, 219 (43.8%) had moderate count, while 111 (22.2%) had high CD4 count (Table 2). Again, out of the 170 participants with low CD4 count, 108 (63.5%) were females, 62 (36.5%) were males, and 55 (32.4%) were within the age of 41–50 years (Table 2). The mean CD4 cell count was 464.4 cells/µL (95%CI 435.8–492.9) (Table 1).

## Comparison of patients' demography and clinical characteristics

Male and female patients were compared in terms of age, CD4 cell count, HIV-1 RNA viral load, and HBV and HCV co-infections to determine association between groups. Males (46.57

**Table 2. Correlation between CD4 count and HIV-1 RNA levels.**

| | CD4+ Cell Count (cells/μL) | | | HIV-1 RNA Level (copies/mL) | | P-value | Viral Suppression | | P-value |
|---|---|---|---|---|---|---|---|---|---|
| | <300 | 300–700 | >700 | Undetectable (<20) | Detectable (>20) | | <1000 | >1000 | |
| Characteristics | N (%) | N (%) | N (%) | N (%) | N (%) | | N (%) | N (%) | |
| Gender | | | | | | 0.2564 | | | |
| Female | 108 (21.6) | 145 (29.0) | 80 (16.0) | 177 (35.4) | 156 (31.2) | | 222 (44.4) | 111 (22.2) | |
| Male | 62 (12.4) | 74 (14.8) | 31 (6.2) | 89 (17.8) | 78 (15.6) | | 114 (22.8) | 53 (10.6) | |
| Total | 170 (34.0) | 219 (43.8) | 111 (22.2) | 266 (53.2) | 234 (46.8) | | 336 (67.2) | 164 (32.8) | |
| Age (years) | | | | | | | | | |
| ≤20 | 5 (1.0) | 15 (3.0) | 4 (0.8) | 11 (2.2) | 13 (2.6) | | 12 (2.4) | 12 (2.4) | |
| 21–30 | 16 (3.2) | 26 (5.2) | 1 (0.2) | 31 (6.2) | 21 (4.2) | | 36 (7.2) | 16 (3.2) | |
| 31–40 | 37 (7.4) | 55 (11.0) | 24 (4.8) | 60 (12.0) | 56 (11.2) | | 75 (15.0) | 41 (8.2) | |
| 41–50 | 55 (11.0) | 64 (12.8) | 35 (7.0) | 84 (16.8) | 70 (14.0) | | 107 (21.4) | 49 (9.8) | |
| 51–60 | 41 (8.2) | 32 (6.4) | 31 (6.2) | 59 (11.8) | 55 (11.0) | | 79 (15.8) | 35 (7.0) | |
| ≥61 | 16 (3.2) | 17 (3.4) | 7 (1.4) | 21 (4.2) | 19 (3.8) | | 27 (5.4) | 11 (2.2) | |
| Average age | 44.5 ± 11.4 | 42.3 ± 12.4 | 44.6 ± 11.3 | 42.6 ± 11.7 | 44.0 ± 11.9 | 0.3027 | | | |
| Total | 170 (34.0) | 219 (43.8) | 111 (22.2) | 266 (53.2) | 234 (46.8) | | 336 (67.2) | 164 (32.8) | |
| Mean CD4 count | 151.9 ± 91.2 | 468.6 ± 107.4 | 934.5 ± 261.6 | 538.7 (95% CI 482.9–594.6) | 365.5 (95% CI 325.4–405.6) | 0.0062 | 541.2 (95% CI 508.5–573.8) | 309.9 (95% CI 261.9–357.9 ) | 0.0053 |
| Mean viral load ±SD (Log10) | 3.70 ± 1.98 | 2.22 ± 1.94 | 1.90 ± 1.81 | | | | | | |

±11.41 years) were older than females (41.01±10.32 years) (Table 3). The mean CD4 cell count for males (435.2 cells/μl [95% CI 387.3–483.1]) was also lower than that of females (479.0 cells/μL [443.5–514.5]) although the difference was not statistically significant ($P$ = .155) (Table 1). However, the mean HIV RNA viral load was $Log_{10}$ 2.82 ± 2.11 copies/mL for females and $Log_{10}$ 2.79 ± 2.05 copies/mL for males ($P$ = .141) (Table 1).

Furthermore, a comparison was made between HIV RNA viral load status and age, CD4 cell count, HBV and HCV status of patients to determine which parameters were associated with HIV RNA viral load status. The study showed that patients with viral suppression were slightly older (43.96±11.93 years) than patients who did not achieve viral suppression (42.56 ±11.7 years) (Table 3). Mean CD4 cell count was significantly ($P$ = .006) higher in patients with undetectable levels of HIV-1 RNA (538.7 cells/μl) than in patients with detectable levels (365.5.9 cells/μl) (Table 2). Mean CD4 counts of patients with viral suppression and patients with virologic failure was 541.2 cells/μL (95% CI 508.5–573.8) and 309.9 cell/μL (95% CI 261.9–357.9) respectively (Table 2). Out of the 42 HIV/HBV co-infected patients, one-third (14/42) had no viral suppression and the only HIV/HCV co-infected patient showed viral suppression (Table 4).

## HIV/HBV and HIV/HCV co-infections

Factors associated with HIV/HBV co-infection were analysed using logistic regression. A bivariate analysis was done to determine the association between variables (age, gender, CD4 count, viral load, and HIV/HBV co-infection) using Chi square test. Associations were considered significant with a *P-value* of 0.05 or less. As shown in Table 4, patients who were HIV/HBV co-infected were predominantly females (27/42 [64.3%]) ($X^2$ [df] = .11 [2]; $P$ value = .740). HIV/HBV co-infection was found in all age groups ($X^2$ [df] = 7.99 [5]; $P$ value = .157) except in participants who were <20 years. The average age for HIV mono-infected patients

**Table 3. Gender and viral suppression according to age difference.**

| | Sex | | HIV Viral Suppression | |
|---|---|---|---|---|
| | **Male** | **Female** | **<1000** | **>1000** |
| **Characteristics** | **N (%)** | **N (%)** | **N (%)** | **N (%)** |
| **Age (years)** | | | | |
| ≤20 | 8 (4.8) | 17 (5.1) | 12 (3.6) | 12 (7.3) |
| 21–30 | 8 (4.8) | 43 (12.9) | 36 (10.7) | 16 (9.8) |
| 31–40 | 24 (14.4) | 92 (27.6) | 75 (22.3) | 41 (25.0) |
| 41–50 | 58 (34.7) | 96 (28.8) | 107 (31.8) | 49 (29.9) |
| 51–60 | 50 (29.9) | 64 (19.3) | 79 (23.5) | 35 (21.3) |
| ≥61 | 19 (11.4) | 21 (6.3) | 27 (8.1) | 11 (6.7) |
| Average age | 46.57±11.4 | 41.04±10.3 | 43.9 ± 11.7 | 42.6 ± 12.5 |
| **Total** | **167 (33.4)** | **333 (66.6)** | **336 (67.2)** | **164 (32.8)** |

**Table 4. Effect of HBV and HCV co-infection on serologic and virologic response.**

| | HIV/HBV & HIV/HCV Co-infections | | | | | | |
|---|---|---|---|---|---|---|---|
| | **HIV mono-infection** | **HIV/HCV co-infection** | | | **HIV/HBV co-infection** | | |
| **Characteristic** | **N (%)** | **N (%)** | **X² (df)** | **P-value** | **N (%)** | **X² (df)** | **P-value** |
| **Sex** | | | | | | | |
| Female | 306 (61.2) | 1 (0.2) | 0.50 (2) | 0.478 | 27 (5.4) | 0.11 (2) | 0.74 |
| Male | 151 (30.2) | 0 (0) | | | 15 (3.0) | | |
| **Total** | **457 (91.4)** | **1 (0.2)** | | | **42 (8.4)** | | |
| **Age (years)** | | | | | | | |
| <20 | 24 (4.8) | 0 (0) | 3.39 (5) | 0.64 | 0 (0) | 7.99 (5) | 0.157 |
| 21–30 | 43 (8.6) | 0 (0) | | | 8 (1.6) | | |
| 31–40 | 104 (20.8) | 0 (0) | | | 12 (2.4) | | |
| 41–50 | 142 (28.4) | 0 (0) | | | 12 (2.4) | | |
| 51–60 | 105 (21.0) | 1 (0.2) | | | 9 (1.8) | | |
| > 60 | 39 (7.8) | 0 (0) | | | 1 (0.2) | | |
| Average age | 43.8 ± 12.0 | 53 | | | 40.9 ±9.8 | | |
| **Total** | **457 (91.4)** | **1 (0.2)** | | | **42 (8.4)** | | |
| **CD4 Count (cells/mL)** | | | | | | | |
| < 300 | 151 (30.2) | 0 (0) | 1.29 (3) | 0.526 | 18 (3.6) | 2.36 (3) | 0.307 |
| 300–700 | 201 (40.2) | 1 (0.2) | | | 18 (3.6) | | |
| >700 | 105 (21.0) | 0 (0) | | | 6 (1.2) | | |
| Mean CD4 Count | 473.8 ± 326.7 | 543.0 ± 325 | | | 361.0 ± 284.0 | | |
| **Total** | **457 (91.4)** | **1 (0.2)** | | | **42 (8.4)** | | |
| **HIV Viral Load (copies/mL)** | | | | | | | |
| Undetectable | 245 (50.8) | 0 (0) | 1.14 (2) | 0.566 | 20 (4.0) | 0.61 (2) | 0.738 |
| Detectable | 212 (42.4) | 1 (0.2) | | | 22 (4.4) | | |
| <1000 | 308 (61.6) | 1 (0.2) | 0.49(1) | 0.484 | 28 (5.6) | 0.006 (2) | 0.939 |
| >1000 | 149 (29.8) | 0 (0) | | | 14 (2.8) | | |
| Mean Viral Load | $Log_{10}$2.8 ± 2.1 | $Log_{10}$1.3 | | | $Log_{10}$2.9 ± 2.0 | 0.057 (1) | 0.811 |
| **Total** | **457 (91.4)** | **1 (0.2)** | | | **42 (8.4)** | | |

Statistics significant at p-value < 0.05

was 43.8 ± 12 years and 40.9 ±9.8 years for HIV/HBV co-infected patients. Mean CD cell count was 361.0 ± 284.0 in HIV/HBV co-infected patients and 473.8 ± 326.7 in HIV mono-infected patients. No significant differences (($X^2$ [df] = 2.36 [3]; *P* value = .307) were observed between HIV/HBV co-infected patients and HIV mono-infected patients. The mean HIV-1 RNA level was not significantly different ($X^2$ [df] = .057 [1]; *P* = .811) among HIV/HBV co-infected patients ($Log_{10}$2.9±2.0 copies/mL), compared to that of HIV mono-infected patients ($Log_{10}$2.8±2.1 copies/mL) although HIV mono-infected patients had lower viral load levels. Among patients with HIV mono-infection, HIV-1 RNA was detectable in 212 (42.4%) and undetectable in 245 (50.8%), whereas in patients with HIV/HBV co-infection HIV-1 RNA was detectable in 22 (4.4%) and undetectable in 20 (4.0%) (Table 4). Out of all 500 HIV-positive patients who took part in the study, only one (a female) had HIV/HCV co-infection with no statistical difference between variables (Table 2).

## Discussion

A total of 500 adult patients between the ages of 18 and 71 were involved in the study. The average age of the participants was 43.56 ±11.87 years. There were higher proportions of females (66.6%) than males (33.4%) generally because of the high proportion of female partici-pants than males, and higher proportions of participants (30.9%) between the ages of 41 and 50 years than any other age group. The gender disparity may also be due to the fact that HIV is 1.62 times more prevalent in adult women than in men [32]. It is thought that females are more biologically susceptible to HIV and AIDS than men [33]. It is also suggested that the risk of acquiring HIV during vaginal sex is higher in women than in men due to the ability of HIV to pass through the vaginal lining and the large surface area of the vagina [34, 35].

From the study, the prevalence of HIV/HBV co-infection among Ghanaian HIV-positive patients was found to be 8.4% [95%CI: 6.2%-11.1%]. A systematic study done in some regions of Ghana previously reported a wide range of HIV/HBV coinfection to be between 2.4 to 41.7% [36] with a pooled HIV/HBV coinfection prevalence rate 13.6%. This was slightly higher than similar studies done in some other countries in Africa (6.7%), Asia (5.9%), Central/South America (5.1%), and North America (4.8%) as reported by Thio et al. [37]. The findings of this study were also consistent with studies in Asian-Pacific Region (10.5%), Cambodia (11.0%), Australia (9.9%), and Nigeria (17.3%) [38–41] that recorded higher prevalence. The majority of the HIV/HBV co-infected patients, (57.1%), were within the age group 31 to 50 years, as was the majority (53.9%) of the overall HIV-positive patients in the study. Although HIV and HBV have shared route of transmission, and, therefore, it was possible for people infected with hepatitis B virus to be infected with HIV and vice versa, it should be mentioned that HBV in Africa is usually acquired early in life perinatally or horizontally whereas HIV is acquired when individuals become sexually active. The study also showed that HIV patients less than 20 years' old had no HIV/HBV co-infection. It is worthy to mention that these patients (≤20 years old) were born or were babies at a time of a massive vaccination campaign against the hepatitis B virus in children in Ghana using the Penta Vaccine in the early 2000s. This could account for the reason they tested negative for HBV. There was a higher proportion (64.3%) of female HIV/HBV co-infected patients than males (35.7%).

The study also found that prevalence of HIV/HCV co-infection was 0.2% [95%CI: 0.0%-0.9%] and this finding was consistent with several studies where HIV/HCV co-infection was found to be lower [42, 43], although some studies have demonstrated higher HIV/HCV co-infection [38–41]. In the CAESAR (Canada, Australia, Europe, South Africa) study, the preva-lence of HIV/HCV co-infection ranged from 1.9% in South Africa to 48.1% in Italy [44]. It is believed that sexual transmission of HCV is relatively inefficient, and the rate of co-infection

among HIV-infected patients with a sexual risk factor is less than 10% [45]. There is a documentary evidence that HIV/HCV co-infections are higher among injecting drug users [46, 47]. In this current study, there were no reports of injecting drug use by study participants, and this may have contributed to the low prevalence of HIV/HCV co-infection among HIV patients recruited for the study. It is also worthy of mentioning that there was no HBV/HCV co-infection among the study participants.

Patient's HIV-1 RNA reaching an undetectable level (<20 copies/mL) is one of the most important goals of anti-retroviral treatment. The purpose of anti-retroviral drugs is to suppress the HIV viral load and increase levels of $CD4^+$ T-cells. Viral suppression is defined as <1000 copies/mL and defines treatment success [48]. According to WHO, viral suppression among PLHIV is one of the 10 global indicators for the health sector response to HIV [49]. In this study, more than two-thirds (67.2%) of patients achieved viral suppression of less than 1000 copies/mL. Although other studies have found that HIV-infected females are younger and tend to have lower HIV-1 RNA levels [36, 50], in this study there was no statistical difference (*P* value = .14) between the mean HIV viral load of female participants ($Log_{10}$ 2.82 ± 2.11 copies/mL) compared to male participants ($Log_{10}$ 2.79 ± 2.05 copies/mL). The study also showed that HIV patients between 31–60 years achieved better virological response, although there was no statistical difference (*P* = .253) between age and viral load levels. All participants self-reported adherence to medications which may have been due to an increased national campaign on the need to strictly adhere to therapy. The strict adherence to treatment regimen may have subsequently contributed to the viral suppression reported.

Out of the 500 participants who took part in the study, 66% (330) had CD4 count greater than 300 cells/μl whereas 34% (170) had CD4 count of less than 300 cells/μl. Although females had slightly higher CD4 count than males (females: 479.0 cells/μL (95% CI 443.5–514.5; males: 435.2 cells/μL (95% CI 387.3–483.1), the study could not establish any gender difference in the immune recovery of patients (*P* value = .939). However, overall treatment outcomes were better in females than in males. The study also demonstrated higher CD4 counts in patients between the ages of 31 and 60 years old. There was, however, no statistical difference (*P* value = .524) between age and CD4 count. Again, CD4 count was significantly higher (*P* = .0062) in patients whose HIV-1 RNA were below levels of detection. The purpose of anti-retroviral drugs is to suppress the HIV viral load and increase proliferation of $CD4^+$ T-cells [51]. As the number of patients with high CD4 counts increased (66%), there was a corresponding increase in patients with viral suppression (67.2%). The study observed a correlation between CD4 counts and HIV viral load levels (*P* value = .0062). Higher CD4 cell counts were associated with lower HIV viral load levels (CD4 count to viral load [$Log_{10}$] ratio—151.9 ± 91.2:3.70 ± 1.98; 468.6 ± 107.4:2.22 ± 1.94; 934.5 ± 261.6:1.90 ± 1.81). This was possible partly due to the fact that higher CD4 count meant better and stronger immune system, resulting in better treatment outcome.

Furthermore, a bivariate analysis using chi square test was used to assess the effect of HIV/HBV co-infection on the immunologic and virologic response of study participants. The study showed that although CD4 cell counts were lower in patients with HIV/HBV co-infection than in patients with HIV mono-infection, there was no statistical difference between the two groups (*P* value = .307). Also, this was similar with patients with HIV/HCV co-infection and patients with HIV mono-infection. This study found no statistical difference (*P* value = .526) on the immunologic recovery between HIV/HCV co-infection and HIV mono-infection. However, in a cohort study of HIV patients in Asia, HIV/HCV co-infection was found to be associated with lower CD4 cell recovery [38]. Also, Lincoln et al. [40] posits that HIV/HCV co-infected patients have poorer response to anti-retrovirals (ARVs) in terms of CD4 count change [39].

Patients on HAART are considered to have HIV-RNA viral load suppression when their HIV viral load falls below 1000 copies/ml. In the study, HIV/HBV co-infected patients had higher HIV-1 viral load ($Log_{10}2.9 \pm 2.0$) compared to HIV mono-infected patients ($Log_{10}2.8 \pm 2.1$) although no significant difference was seen (*P* value = .811). This finding agrees with several independent studies conducted elsewhere [38, 41, 44], that found the effect of HIV/HBV co-infection and HIV mono-infection on HIV viral load of patients not to be statistically significant. However, a study conducted by Yang et al., [52] found patients with HIV/HBV co-infection to be less likely to achieve HIV RNA suppression and CD4 increase. HIV-1 viral load of the only HIV/HCV co-infected patient appeared to be higher ($Log_{10}1.3$) than HIV-1 viral load of HIV mono-infected patients. This was expected partly due to the fact that only 1/500 patients had HIV/HCV co-infection (0.2%) among the participants. This finding was also consistent with the findings of other studies [38, 41] in which HCV infection in HIV patients had no effect on HIV RNA suppression.

## Conclusion

This study showed a prevalence of 8.4% and 0.2% HIV/HBV and HIV/HCV co-infection among clinically diagnosed HIV-1 positive patients respectively. There was no HBV/HCV co-infection in HIV patients on treatment. The study also showed no significant difference in the immunologic and virologic responses of patients with HIV mono-infections had and patients with HIV/HBV and HIV/HCV coinfections, and that treatment outcomes were better in females than in males. The study, therefore, concludes that, HIV/HBV and HIV/HCV coinfections do not significantly affect the immunologic and virologic responses of patients who have initiated highly active antiretroviral therapy. The study, however, recommends the use of ARVs active against HBV (such as lamivudine (3TC) tenofovir disoproxil fumarate (TDF) and tenofovir alafenamide (TAF)), in ART regimens for HIV co-infected patients for better treatment outcomes of patients co-infected with HIV and HBV.

## Supporting information

**S1 Data.**
(XLS)

## Acknowledgments

The authors would like to appreciate Mr. Benjamin Nartey Amanya and Moses Adongo for their immense support and assistance. Our appreciation also goes to the entire clinical staff of the Fevers Unit of the Korle-Bu Teaching hospital.

## Author Contributions

**Conceptualization:** Lawrence Annison, Samuel Antwi-Baffour.

**Data curation:** Lawrence Annison, Paulina Franklin Eshun, Peter Forson.

**Formal analysis:** Sharon Annison.

**Funding acquisition:** Lawrence Annison, Henry Hackman, Samuel Antwi-Baffour.

**Investigation:** Lawrence Annison, Paulina Franklin Eshun, Peter Forson.

**Methodology:** Lawrence Annison.

**Project administration:** Lawrence Annison.

**Resources:** Lawrence Annison, Henry Hackman, Samuel Antwi-Baffour.

**Software:** Sharon Annison.

**Supervision:** Lawrence Annison, Henry Hackman, Peter Forson.

**Validation:** Lawrence Annison, Samuel Antwi-Baffour.

**Writing – original draft:** Lawrence Annison.

**Writing – review & editing:** Henry Hackman, Samuel Antwi-Baffour.

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
