## [Decision Letter · Decision Letter 0]

12 Sep 2022

PONE-D-22-19884SEROPREVALENCE AND EFFECT OF HBV AND HCV CO-INFECTIONS ON THE IMMUNO-VIROLOGIC RESPONSES OF ADULT HIV-INFECTED PERSONS ON ANTI-RETROVIRAL THERAPY

PLOS ONE

Dear Dr. Annison,

Thank you for submitting your manuscript to PLOS ONE. After careful consideration, we feel that it has merit but does not fully meet PLOS ONE’s publication criteria as it currently stands. Therefore, we invite you to submit a revised version of the manuscript that addresses the points raised during the review process.

The manuscript has been evaluated by one reviewer, and their comments are available below.

The reviewer has raised a number of  concerns regarding the reporting, methodology and statistical analysis of the study. 

Could you please revise the manuscript to carefully address the concerns raised?

Please note that we have only been able to secure a single reviewer to assess your manuscript. We are issuing a decision on your manuscript at this point to prevent further delays in the evaluation of your manuscript. Please be aware that the editor who handles your revised manuscript might find it necessary to invite additional reviewers to assess this work once the revised manuscript is submitted. However, we will aim to proceed on the basis of this single review if possible.

We look forward to receiving your revised manuscript.

Kind regards,

Johannes Stortz

Staff Editor

PLOS ONE

Journal Requirements:

3. Please amend your authorship list in your manuscript file to include authors Peter Forson and Sharon Annison.

Reviewers' comments:

Reviewer's Responses to Questions

**Comments to the Author**

1. Is the manuscript technically sound, and do the data support the conclusions?

Reviewer #1: No

2. Has the statistical analysis been performed appropriately and rigorously? 

Reviewer #1: Yes

3. Have the authors made all data underlying the findings in their manuscript fully available?

Reviewer #1: No

4. Is the manuscript presented in an intelligible fashion and written in standard English?

Reviewer #1: Yes

5. Review Comments to the Author

Reviewer #1: The study sought to determine the seroprevalence of HIV/HBV and HIV/HCV co- infections among HIV patients on antiretroviral therapy and to evaluate the effect of HIV/HBV and HIV/HCV co-infections on the immunologic and virologic responses of patients. The effects of coinfections in HIV is an important topic and fills a gap especially in sub Saharan Africa. However, the paper needs some revisions.

1. The population size was not provided and important results not mentioned in the abstract. “HIV/HBV and HIV/HCV co-infections have varied effects on the immunological and virological response of HIV patients on ART” is not adequate for results section. Because of this the conclusions had results which were not mentioned in the results section. Yet conclusions are supposed to be supported by results.

2. The conclusion “Patients with HIV mono-infections are more likely to have sustained HIV viral suppression and higher CD4 cell count, and that HIV/HBV and HIV/HCV coinfections affect the immunologic and virologic response of patients who have initiated highly active antiretroviral therapy” is not supported by results as the associations in table 4 were not statistically significant.

3. PLHIV not defined at first mention

4. The sentence in the introduction about no sero-epidemiology data in HBV and HCV in sub-Saharan Africa need to be qualified. Does it refer to national data? The paper referenced is for a single study not a review paper or a synthesis paper on available data.

5. Check references eg The tittle for reference 23 has typos.

6. The paper is citing 2010 data for HIV and HBV prevalence in sub-Saharan Africa (22.5 and 52 million) while current epidemiology data is available for the 2 pathogens.

7. HBsAg was defined several times in the paper

8. Manufactures details were not provided for the Advanced Quality One Step Tests.

9. Include confidence intervals for prevalence rates and just report the prevalence for the reactive results. The non reactive results are implied. Similar to male or female.

10. For statistical analysis section include the p value significance cut off.

11. Reference of WHO guidelines used for HIV RNA categories was not provided

12. Include percentages here “Patients with viral load below undetectable levels were made up of 177/266 females and 89/266 males”.

13. Were there any HBV/HIV or HIV/HCV participants with virologic failure?

14. Table 7 referred here might be a typo “……………detectable in 28 (66.7%) and undetectable in 14 (33.3%) (Table 7)”

15. The prevalence rates should also be compared to previous studies from Ghana

16. Isn’t this “There was a higher proportion (64.3%) of female HIV/HBV co-infected patients than males (35.7%)” because there were more females in the study in general? Eg There were higher proportions of females (66.6%) than males (33.4%)

17. In the discussion it is stated that “in this study the mean HIV viral load of female participants was slightly higher but the difference was not statistically significant. The sentence in the discussion about HIV RNA appearing to be higher is misleading in the discussion is the difference was not statistically different

18. The study concluded that HIV/HBV coinfection was low but 8.4% is not necessarily low. Also check the interpretation of non significant results in the conclusion. Which HCV treatment would be included as part of the ART regimen in reference to this sentence “The study therefore recommends the use of ARVs active against HBV and HCV in ART regimens for HIV co-infected patients regardless of the CD4 cell count level”. Also considering the fact that there was only one patient with HCV in this study.

Data Availability

The paper states that “Data can be made available upon a reasonable request” which might restrict the availability as some requests might be deemed unreasonable.

6. PLOS authors have the option to publish the peer review history of their article (what does this mean?). If published, this will include your full peer review and any attached files.

Reviewer #1: No

---

## [Author Response · Author response to Decision Letter 0]

2 Oct 2022

Medical Laboratory Technology Dept.,

Accra Technical University

P. O. Box GP 156,

Accra, Ghana.

1st October 2022

Dr. Johannes Stortz

Staff Editor

PLOS ONE

Dear Dr. Stortz,

Re: Resubmission of manuscript reference no. PONE-D-22-19884.

Please find attached a revised version of our manuscript originally titled “Seroprevalence and effect of HBV and HCV co-infections on the immuno-virologic responses of adult HIV-infected persons on anti-retroviral therapy”, which was submitted for consideration for publication in the PLOS ONE Journal.

We wish to thank you and the reviewers for your insightful comment. These have helped us significantly improve the quality of our manuscript.

In accordance with your comments and that of the Reviewer, revisions to our manuscript have been done using Track Changes in Microsoft Word. The manuscript has been formatted to meet PLOS ONE style requirements, and the authorship list has been amended to include Peter Forson and Sharon Annison. Our point-by-point responses to the Reviewer’s comments are highlighted in yellow. We hope that the revisions in the manuscript and our accompanying responses will be sufficient to make our manuscript suitable for publication in the PLOS ONE Journal. 

1. The population size was not provided and important results were not mentioned in the abstract.

Population size has been included accordingly. Results have been included in the Abstract accordingly.

2. The conclusion “Patients with HIV mono-infections are more likely to have sustained HIV viral suppression and higher CD4 cell count, and that HIV/HBV and HIV/HCV coinfections affect the immunologic and virologic response of patients who have initiated highly active antiretroviral therapy” is not supported by results as the associations in table 4 were not statistically significant.

Viral suppression and CD4 cell counts were higher in HIV mono-infected individuals in all instances, howbeit they were not statistically significant. Accordingly, the statement has been amended to include “…though there were no statistical difference” 

3. PLHIV not defined at first mention

PLHIV has been defined appropriately 

4. The sentence in the introduction about no sero-epidemiology data in HBV and HCV in sub-Saharan Africa need to be qualified. Does it refer to national data? The paper referenced is for a single study not a review paper or a synthesis paper on available data.

The sentence has been amended.

5. Check references eg The tittle for reference 23 has typos.

Error has been corrected.

6. The paper is citing 2010 data for HIV and HBV prevalence in sub-Saharan Africa (22.5 and 52 million) while current epidemiology data is available for the 2 pathogens.

Current data (2021 for HIV and 2019 for HBV) have been added, and old data deleted.

7. HBsAg was defined several times in the paper

The repeated definition has been deleted.

8. Manufactures details were not provided for the Advanced Quality One Step Tests.

Manufacturer’s details have been added accrodingly.

9. Include confidence intervals for prevalence rates and just report the prevalence for the reactive results. The non reactive results are implied. Similar to male or female.

Implied results have been deleted accordingly, and confidence intervals have been added to our study’s prevalence.

10. For statistical analysis section include the p value significance cut off.

P value cut-off has been included.

11. Reference of WHO guidelines used for HIV RNA categories was not provided

Reference has been provided.

12. Include percentages here “Patients with viral load below undetectable levels were made up of 177/266 females and 89/266 males”.

Percentages have been added appropriately.

13. Were there any HBV/HIV or HIV/HCV participants with virologic failure?

Results for HBV/HIV and HCV/HIV viral suppression can be found in Paragraph 2 line 20 under Results of Comparison of patients’ demography and clinical characteristics, and it has been highlighted in yellow for easy verification. It is also shown in Table 4. 

14. Table 7 referred here might be a typo “……………detectable in 28 (66.7%) and undetectable in 14 (33.3%) (Table 7)”

It is a typographical error and has been corrected. It is Table 4.

15. The prevalence rates should also be compared to previous studies from Ghana

Previous studies in Ghana has been compared with current study.

16. Isn’t this “There was a higher proportion (64.3%) of female HIV/HBV co-infected patients than males (35.7%)” because there were more females in the study in general? Eg There were higher proportions of females (66.6%) than males (33.4%)

That is the case. There were more female participants than males, and that has been stated accordingly. In addition, other reasons that may have accounted for the gender disparity have also been stated. 

17. 17. In the discussion it is stated that “in this study the mean HIV viral load of female participants was slightly higher but the difference was not statistically significant. The sentence in the discussion about HIV RNA appearing to be higher is misleading in the discussion is the difference was not statistically different

The statement has been amended for clarity. 

18. The study concluded that HIV/HBV coinfection was low but 8.4% is not necessarily low. Also check the interpretation of non significant results in the conclusion. Which HCV treatment would be included as part of the ART regimen in reference to this sentence “The study therefore recommends the use of ARVs active against HBV and HCV in ART regimens for HIV co-infected patients regardless of the CD4 cell count level”. Also considering the fact that there was only one patient with HCV in this study.

The authors agree with the Reviewer that 8.4% is not necessarily low. We generalized the HIV/HBV and HIV/HCV together due to the low (0.2%) prevalence of HIV/HCV. The sentence has been deleted accordingly. Approved ARVs active against HBV have been added.

Thank you for your consideration. We look forward to hearing from you.

Sincerely,

Lawrence Annison

(lannison@atu.edu.gh)

---

## [Decision Letter · Decision Letter 1]

24 Oct 2022

PONE-D-22-19884R1SEROPREVALENCE AND EFFECT OF HBV AND HCV CO-INFECTIONS ON THE IMMUNO-VIROLOGIC RESPONSES OF ADULT HIV-INFECTED PERSONS ON ANTI-RETROVIRAL THERAPYPLOS ONE

Dear Dr. Annison,

Thank you for submitting your manuscript to PLOS ONE. After careful consideration, we feel that it has merit but does not fully meet PLOS ONE’s publication criteria as it currently stands. Therefore, we invite you to submit a revised version of the manuscript that addresses the points raised during the review process.

Your revised manuscript was reviewed by one expert in the field. The reviewer indicated that one of the comments was not satisfactory addressed and your conclusions are still not supported by the data. Please respond properly to this comment.

We look forward to receiving your revised manuscript.

Kind regards,

Yury E Khudyakov, PhD

Academic Editor

PLOS ONE

Journal Requirements:

Reviewers' comments:

Reviewer's Responses to Questions

**Comments to the Author**

1. If the authors have adequately addressed your comments raised in a previous round of review and you feel that this manuscript is now acceptable for publication, you may indicate that here to bypass the “Comments to the Author” section, enter your conflict of interest statement in the “Confidential to Editor” section, and submit your "Accept" recommendation.

Reviewer #1: (No Response)

2. Is the manuscript technically sound, and do the data support the conclusions?

Reviewer #1: Partly

3. Has the statistical analysis been performed appropriately and rigorously? 

Reviewer #1: I Don't Know

4. Have the authors made all data underlying the findings in their manuscript fully available?

Reviewer #1: Yes

5. Is the manuscript presented in an intelligible fashion and written in standard English?

Reviewer #1: Yes

6. Review Comments to the Author

Reviewer #1: Comment number 2 is still not addressed. The conclusion “Patients with HIV mono-infections are more likely to have sustained HIV viral suppression and higher CD4 cell count, and that HIV/HBV and HIV/HCV coinfections affect the immunologic and virologic response of patients who have initiated highly active antiretroviral therapy” is not supported by results as the associations in table 4 were not statistically significant.

7. PLOS authors have the option to publish the peer review history of their article (what does this mean?). If published, this will include your full peer review and any attached files.

Reviewer #1: No

---

## [Author Response · Author response to Decision Letter 1]

4 Nov 2022

We wish to thank you and the reviewers for your insightful comment. These have helped us significantly improve the quality of our manuscript.

In accordance with your comments and that of the Reviewer, revisions to our manuscript have been done using Track Changes in Microsoft Word. The Reviewer raised concerns about the conclusions not being supported by the data provided. Our response to the Reviewer’s comment in point 6 is highlighted in yellow. We hope that the revisions in the manuscript and our accompanying response will be sufficient to make our manuscript suitable for publication in the PLOS ONE Journal. 

Academic Editor’s comments: 

Your revised manuscript was reviewed by one expert in the field. The reviewer indicated that one of the comments was not satisfactorily addressed and your conclusions are still not supported by the data. Please respond properly to this comment.

Comments to the Author

1. If the authors have adequately addressed your comments raised in a previous round of review and you feel that this manuscript is now acceptable for publication, you may indicate that here to bypass the “Comments to the Author” section, enter your conflict of interest statement in the “Confidential to Editor” section, and submit your "Accept" recommendation.

Reviewer #1: (No Response)

2. Is the manuscript technically sound, and do the data support the conclusions?

Reviewer #1: Partly

3. Has the statistical analysis been performed appropriately and rigorously?

Reviewer #1: I Don't Know

4. Have the authors made all data underlying the findings in their manuscript fully available?

Reviewer #1: Yes

5. Is the manuscript presented in an intelligible fashion and written in standard English?

Reviewer #1: Yes

6. Review Comments to the Author

Reviewer #1: Comment number 2 is still not addressed. The conclusion “Patients with HIV mono-infections are more likely to have sustained HIV viral suppression and higher CD4 cell count, and that HIV/HBV and HIV/HCV coinfections affect the immunologic and virologic response of patients who have initiated highly active antiretroviral therapy” is not supported by results as the associations in table 4 were not statistically significant.

The authors agree with the Reviewer that associations in Table 4 were not statistically significant. Conclusions have been revised accordingly in the Abstract and Conclusion sessions to reflect the data as captured in Table 4. The revision can be found in the Track Changes attached. 

Thank you for your consideration. We look forward to hearing from you.

Sincerely,

Lawrence Annison

(lannison@atu.edu.gh)

---

## [Editor Report · Decision Letter 2]

9 Nov 2022

SEROPREVALENCE AND EFFECT OF HBV AND HCV CO-INFECTIONS ON THE IMMUNO-VIROLOGIC RESPONSES OF ADULT HIV-INFECTED PERSONS ON ANTI-RETROVIRAL THERAPY

PONE-D-22-19884R2

Dear Dr. Annison,

We’re pleased to inform you that your manuscript has been judged scientifically suitable for publication and will be formally accepted for publication once it meets all outstanding technical requirements.

Kind regards,

Yury E Khudyakov, PhD

Academic Editor

PLOS ONE
---

## [Editor Report · Acceptance letter]

15 Nov 2022

PONE-D-22-19884R2 

Seroprevalence and effect of HBV and HCV co-infections on the immuno-virologic responses of adult HIV-infected persons on anti-retroviral therapy 

Dear Dr. Annison:

I'm pleased to inform you that your manuscript has been deemed suitable for publication in PLOS ONE. Congratulations! Your manuscript is now with our production department. 

Kind regards, 

on behalf of

Dr. Yury E Khudyakov 

Academic Editor

PLOS ONE